# The Clinical Utility of Selected Coagulation Parameters in Predicting the Risk of Venous Thromboembolism in Neuroendocrine Tumours: A Prospective, Single-Centre Study

**DOI:** 10.3390/cancers17213405

**Published:** 2025-10-22

**Authors:** Monika Wójcik-Giertuga, Anna Malczewska-Herman, Arkadiusz Orzeł, Beata Kos-Kudła

**Affiliations:** 1Department of Endocrinology and Neuroendocrine Tumours, Department of Pathophysiology and Endocrinology, Faculty of Medical Sciences in Zabrze, Medical University of Silesia, Katowice, ul. Ceglana 35, 40-514 Katowice, Poland; anna.malczewska@sum.edu.pl (A.M.-H.); bkoskudla@sum.edu.pl (B.K.-K.); 2The University Clinical Center, Medical University of Silesia, Katowice, ul. Ceglana 35, 40-514 Katowice, Poland; arkadiusz.orzel@scanmed.pl

**Keywords:** venous thromboembolism, coagulation parameters, D-dimer, fibrinogen, platelets, antithrombin III, tissue factor, neuroendocrine tumours, tumour thrombosis, survival

## Abstract

Neuroendocrine tumours (NETs) belong to a heterogeneous group of malignant neoplasms that differ in their ability to produce hormones and biogenic amines, which is often associated with a more favourable prognosis for well-differentiated tumours. Despite numerous reports in the literature documenting venous thromboembolism (VTE) events in these patients, data on thromboembolic complications in NETs remain limited, and no specific recommendations exist regarding the use of antithrombotic prophylaxis in this group. Thrombotic risk assessment models have not yet been validated in NETs. This article presents the first prospective analysis of selected coagulation parameters, the incidence and risk factors for VTE, and evaluates thromboembolic risk assessment models as well as survival data in pancreatic and small intestinal NETs. Our aim was to improve patient stratification for VTE risk and to help identify patients with NETs who might benefit from antithrombotic prophylaxis.

## 1. Introduction

The incidence of venous thromboembolism (VTE) markedly increases mortality in cancer patients, potentially by as much as fourfold [1]. There are various potential mechanisms related to cancer-associated thrombosis. Virchow’s triad highlights three key risk factors that may contribute to the development of thrombosis, including damage to the vessel wall, stasis of the blood flow, and hypercoagulability [2]. VTE may present as deep vein thrombosis (DVT), pulmonary embolism (PE), or atypical thrombosis, which may include abdominal tumour thrombosis (TT) such as portal, splanchnic, and mesenteric vein thrombosis. Superficial venous thrombosis (SVT), regarded as a benign condition, may also lead to the development of DVT and/or PE [3]. Effective blood biomarkers indicating the development of VTE in cancer patients are needed. Candidate biomarkers include blood levels of D-dimer (DD), fibrinogen, platelets (PLT), antithrombin III (AT-III), and tissue factor (TF).

Well-differentiated neuroendocrine tumours (NETs) are biologically and clinically distinct from other malignancies because of their ability to produce hormones in the case of functioning NETs (F-NETs), as well as a typically msore favourable prognosis and a different treatment approach due to the presence of somatostatin receptors on their surfaces [4]. NETs can also exhibit an increased risk of VTE [5], but the mechanisms behind this are still not well understood. It is assumed that some angiogenic factors, such as vascular endothelial growth factor (VEGF), may be good candidates to be prognostic factors of VTE [6,7]. NET cases also presented a high expression of proangiogenic factors [8,9], but the link between VTE and these parameters warrants further research. Additionally, the excessive serotonin production in F-NETs can also contribute to the damage of the vascular endothelium and lead to a loss of anticoagulant properties, facilitating clot formation [10]. However, this requires broadening our knowledge in this area. Despite a large number of reported cases of VTE in NETs [11,12], there are currently no guidelines on the assessment of coagulation parameters or the application of thromboembolic risk assessment models in this group. It is assumed that conventional VTE risk assessment models, such as the Khorana score (KS) and Vienna-CATS score (VC-S), may not apply to NETs, as they were developed and validated on a different group of oncological patients, often with completely different biology and aggressiveness [13]. However, the appropriate assessment of the thromboembolic risk remains vital to effectively identify patients at risk of VTE to implement preventive management such as thromboprophylaxis. The primary objectives of this work were to assess the predictive validity of various VTE risk assessment scales in predicting VTE risk in patients with pancreatic (Pan-NETs) and small intestine (SI-NETs) NETs, and the secondary objective was to determine the clinical utility of blood coagulation biomarkers as indicators of VTE in these patients.

## 2. Materials and Methods

All patients were enrolled at the Department of Endocrinology and Neuroendocrine Tumours, ENETS Center of Excellence, Medical University of Silesia, in Katowice, Poland. Informed written consent was obtained from all study participants. The study was conducted in accordance with the Declaration of Helsinki and Good Clinical Practice guidelines. The Ethics Committee of the Medical University of Silesia approved the study protocol (approval number: BNW/NWN/0052/KB1/147/I/22/23). The inclusion and exclusion criteria for the study are shown in Table 1.

Blood levels of selected parameters, including DD, fibrinogen, PLT, AT-III, and TF, were measured. These parameters were then correlated with four VTE risk assessment scales (the Khorana score [KS], the Vienna CATS score [VC-S], the ONCOTEV score [ONCO-S], and the Padua Prediction Score [PPS], Table 2). The KS and the VC-S are used to predict the risk of VTE in cancer patients, especially those receiving anticancer therapy [14,15]. The ONKO-S can also serve as a VTE risk prediction model for cancer patients to help stratify VTE risk and identify a higher occurrence of VTE [16]. The PPS is employed to evaluate the risk of VTE in hospitalized, non-surgical patients [17]. The coagulation parameters were also correlated with other variables such as survival data, ECOG/WHO scale, primary tumour site, clinical staging, secretory status, concentration of selected NET markers (chromogranin A, serotonin, and urinary 5-hydroxyindoleacetic acid), disease progression (based on radiological evaluation), and treatment history. All study participants underwent screening for DVT or SVT using a venous Doppler ultrasound of the lower extremities and iliac vessels, including a B-mode compression test and a Doppler examination. A qualified specialist in angiology performed all examinations. Patients were followed up for VTE events for 12 months. The overall survival (OS) was calculated as the time from blood sample collection to either the patient’s death or the end of follow-up.

### 2.1. Blood Sampling and Laboratory Analysis

Venous blood samples were collected from all study participants. The testing tubes contained trisodium citrate. The ACL TOP 350 analyzer was used to assess plasma levels of DD, fibrinogen, and AT-III, and the Sysmex XN-1000 analyzer was used to measure the blood count. Plasma levels of TF were determined using enzyme-linked immunosorbent assay (ELISA).

### 2.2. Statistical Analysis

Data analysis and visualization were conducted using R version 4.5.0 (The R Foundation for Statistical Computing, Vienna, Austria) in R Studio version 2024.12.1 build 563 (PBC, Boston, MA, USA). Kruskal–Wallis and Fisher-exact tests were used for group comparisons. The Spearman rank correlation coefficient (rS) was used to correlate the biochemical, clinical, and demographic parameters using the stats package. Cox proportional hazards models and Kaplan–Meier survival curves were generated using R with the survival package, survminer package, and dplyr package. Continuous parameters were converted to categorical variables by dividing the dataset into 5 groups, where each group contained 20% of the cases, ordered from the lowest to the highest values (labelled very low, low, medium, high, and very high) due to Kaplan–Meier analysis requirements. All the data were presented as numbers of cases (n) and mean values ± standard deviation (SD). The value of *p* < 0.05 was considered statistically significant.

### 2.3. Multivariate Survival Analysis

Given the limited sample size, multivariate survival analyses were performed on selected parameters, chosen based on the study’s scope, preliminary survival results, and statistical analyses (including correlations with survival and other sociodemographic and biochemical parameters). Age and metastasis did not demonstrate a statistically significant association with patient survival, whereas the impact of clinical staging was consistent with findings reported in previous studies. Notably, clinical staging differed between tumour locations (Pan-NETs vs. SI-NETs) in our study; therefore, in the Cox proportional hazards analysis, clinical staging was excluded in favour of tumour location. For consistency, the results of the Cox proportional hazards model in this article are presented with continuous variables categorized in the same manner as in the Kaplan–Meier analysis. Results of the Cox model for continuous variables are available in the Appendix A.

### 2.4. Missing Data Handling

A review of the dataset in the study group showed no missing values for sociodemographic parameters (age and sex), body mass index (BMI), selected coagulation parameters (DD, fibrinogen, PLT count, and AT-III), survival data, ECOG/WHO scale, primary tumour site, clinical staging, grading, secretory status, concentration of selected NET markers (chromogranin A, serotonin, urinary 5-hydroxyindoleacetic acid), cortisol, disease progression, and treatment and VTE risk assessment scales. The only missing data pertained to TF, where levels for 5 patients (5.05%) were undetermined. Given that this represented a small portion of the total sample, a complete case analysis was deemed appropriate for survival modelling. For all other statistical analyses that did not involve the TF variable, data from the full cohort of patients were included.

No missing data for measured parameters were reported in the control group.

## 3. Results

The study group included 99 NET patients, of which Pan-NETs comprised 63.6% (*n* = 63) and 36.4% of SI-NETs (*n* = 36). The control group included 47 healthy volunteers. No significant differences in age, sex, or BMI between the study and control groups. The follow-up for VTE events ranged due to differences in the timing of patient inclusion during the over 1.5-year recruitment phase but was approximately 12 months. The clinicopathological characteristics of both the study and control groups are shown in Table 3.

### 3.1. The Occurrence of VTE Events and VTE Risk Assessment Scales

VTE events were observed at the start of the study, before the follow-up period, in 10% of patients (*n* = 10). These included DVT (*n* = 1), both DVT and SVT (*n* = 1), SVT alone (*n* = 2), and TT (*n* = 6), which comprised splanchnic vein thrombosis (*n* = 5) and portal vein thrombosis (*n* = 1) (Table 4). None of the patients exhibited PE symptoms, nor was this condition detected during follow-up imaging (CT scans) (Figure 1). No VTE events were observed in the control group. The assessment of study and control groups according to the VTE risk assessment scales is presented in Table 3. 

### 3.2. Selected Blood Parameters of Coagulation

#### 3.2.1. D-Dimer

The mean plasma concentration of DD was significantly higher in the study group compared to the control group (957.59 ± 2021.86 vs. 400.26 ± 230.55 µg/L, *p* = 0.007) (Table 5), but no differences between Pan-NETs and SI-NETs were observed (Table 6). The highest DD levels were noted in patients with liver and bone metastases (1717 ± 3151.1 vs. 2073.4 ± 2288.9 µg/L, respectively), and in patients with progressive disease compared to stable disease (2513.7 ± 3624.3 vs. 431.9 ± 244.7 µg/L, *p* ≤ 0.001). DD levels were statistically higher in patients who passed away compared to those who remained alive at the last follow-up (1749.2 ± 445.2 vs. 915.5 ± 2064.9 µg/L, *p* = 0.002). No statistical differences were observed in DD concentrations depending on the treatment used.

#### 3.2.2. Fibrinogen

The mean plasma concentration of fibrinogen showed no difference between the study and control groups (318.98 ± 78.74 vs. 303.40 ± 55.45 mg/dL, *p* = 0.301, respectively) (Table 5), and no differences between Pan-NETs and SI-NETs (Table 6). Patients with progressive disease had higher fibrinogen levels compared to those with stable disease (359.3 ± 106.3 vs. 305.4 ± 62.2 mg/dL, *p* = 0.041). Similarly, patients who died, compared to those who remained alive at the last follow-up, also exhibited higher fibrinogen levels (421.4 ± 123 vs. 313.5 ± 72.7 mg/dL, *p* = 0.025). No statistical differences were observed in fibrinogen levels depending on the treatment used.

#### 3.2.3. Platelets

The mean PLT count was similar between the study and control groups (255.28 ± 97.22 vs. 256.11 ± 54.93, *p* = 0.435) (Table 5), and no differences between Pan-NETs and SI-NETs were observed (Table 6). Patients with higher serotonin levels had higher PLT levels (282.9 ± 99.4 vs. 238.1 ± 100.4, *p* = 0.004). No statistical differences were observed in PLT count depending on the treatment used.

#### 3.2.4. Antithrombin III Activity

The average activity of AT-III was similar in both the study and control groups (101.71 ± 14.95 vs. 102.47 ± 11.34%, *p* = 0.795) (Table 5) and was higher in Pan-NETs compared to Si-NETs (104.63 ± 14.56 vs. 96.58 ± 14.42%, *p* = 0.010) (Table 6). Moreover, patients with TT exhibited higher AT-III levels than those without TT (122 ± 17.1 vs. 100.4 ± 13.9%, *p* = 0.011). Patients with metastatic disease had lower AT-III levels compared to those without metastases (98.3 ± 16.7 vs. 106.4 ± 10.7%, *p* = 0.007). Patients with higher serotonin levels had lower AT-III levels (96.8 ± 14.1 vs. 104.7 ± 14.8%, *p* = 0.013). The lower AT-III levels were noted in patients during somatostatin analogue treatment compared to those without this type of treatment (95.1 ± 15.6 vs. 105.2 ± 13.5%, *p* = 0.003). No statistical differences were observed in AT-III depending on the other treatment used.

#### 3.2.5. Tissue Factor

The mean concentration of TF was similar in both the study and control groups (157.71 ± 63.96 vs. 161.77 ± 25.56 pg/mL, *p* = 0.160) (Table 5) or between Pan-NETs and SI-NETs (Table 6). Patients with localized disease had lower levels of TF than those with metastases (143.8 ± 71.2 vs. 167.2 ± 57.3 pg/mL, *p* = 0.023). No statistical differences were observed in TF levels depending on the treatment used.

The correlations between the measured coagulation parameters and selected clinical parameters are presented in Table 7.

### 3.3. Survival Analysis and VTE Risk Assessment Scales

Preliminary survival analysis with a median follow-up of 12 months uncovered that 5.1% of patients passed away (*n* = 5). All deceased patients were at the IV clinical stage. The cause of death was unknown. The OS was calculated at 94.9%. The selected parameters, which negatively affected survival in NET patients, are presented in Figure 2. Multivariate Cox regression analysis of selected parameters and VTE risk assessment scales was presented in Figure 3. None of the VTE risk assessment scales provided a good measure (AUC > 0.7) in ROC analysis (Table 8, Figure 4). A higher score on the Khorana or Vienna CATS scale compared to other scales was linked to worse survival (*p* = 0.009 and *p* ≤ 0.001, respectively) (Figure 5).

(a)Patients with SI-NETs compared to Pan-NETs exhibited poorer survival (*p* = 0.008).(b)Patients with carcinoid syndrome, compared to NF-NETs, exhibited poorer survival (*p* = 0.005).(c)Increased platelet levels were found to negatively affect survival (*p* = 0.005).(d)Increased TF levels were found to negatively affect survival (*p* ≤ 0.004).(e)Elevated cortisol levels were associated with worse outcomes compared to those with normal levels of cortisol (*p *= 0.029).(f)Additionally, elevated serotonin levels were associated with worse outcomes compared to those with normal levels of serotonin (*p* = 0.003).(g)Increased TF levels were found to negatively affect survival (*p* ≤ 0.001).

## 4. Discussion

The most commonly predisposed cancer sites associated with VTE include the digestive system (in particular, the pancreas and stomach) [18,19,20]. NETs were also reported to present an increase in thrombotic risk [11,12]. To the best of our knowledge, the current study is the first to analyze thromboembolic risk using VTE risk assessment scales and selected coagulation parameters in patients with NETs. According to Massironi et al., in a retrospective cohort of 160 GEP-NET patients, 12 patients developed VTE, and an elevated risk of VTE was found to be associated with the primary tumour site location (particularly in the pancreas) or the higher tumour grade [5]. We also confirmed the most common VTE events in Pan-NETs, and presented that routinely available tests, such as DD and fibrinogen assessment, could help improve risk stratification of VTE in NET cases. Typically, VTE in NETs may occur as DVT and PE. However, NET patients may also develop thrombosis in atypical locations, such as abdominal veins [11]. Nearly 33% of Pan-NETs were found to develop venous tumour thrombi [21]. Almost 33% of non-functioning NETs (NF-NETs), that are typically slow-growing and late presenting tumours, were diagnosed with venous thrombosis within the tumour [22]. TT is common in advanced Pan-NETs [23], particularly at T3 and T4 stages [24,25]. TT was associated with an unfavourable prognosis [26], similar to our findings. In metastatic pancreatic cancer, the occurrence of TT was associated with almost a three-fold higher risk of mortality [26]. Of note, diagnosis of the TT, acute symptomatic or incidentally detected splanchnic vein thrombosis, requires anticoagulation treatment for usually 3–6 months (or sometimes longer) [26].

Plasma levels of DD and fibrinogen are frequently used as indicators of hypercoagulability [27,28]. DD concentration exceeding the upper limit of normal by two-fold correlated with a higher risk of VTE in cancer patients [18,29], and similarly, we observed elevated levels of DD in most cases of VTE events in our study. However, DD remains a non-specific biomarker that can rise due to various causes other than VTE [18]. We also showed, similarly to many different malignancies [28,30,31,32,33,34], that DD and fibrinogen correlated with disease progression in NET patients. Cancer cells are reported to possess potent procoagulant properties that can activate the coagulation system by local activation of thrombin or by secretion of inflammatory factors [35] and increase plasma DD and fibrinogen levels [36]. In the literature, a correlation between DD and VEGF concentration has been noted [37,38]. It is known that VEGF is recognized as a crucial factor that promotes angiogenesis, and this process may assist the migration of cancer cells, increasing blood and oxygen supply to the tumour, and facilitating disease progression [39,40]. The role of fibrinogen in promoting metastasis remains unclear, potentially being the crucial mediator in establishing the tumour microenvironment [41]. Palumbo et al. noticed that fibrinogen is considered an essential determinant of metastasis of cancer cells [42]. In a study on cell lines, fibrinogen administration was shown to induce the expression of adhesion molecules such as ICAM-1 and enhance tumour cell migration, as well as increase angiogenesis and vascular endothelial permeability [43], but further research in NETs is needed.

Thrombocytosis is often observed in cancers of the gastrointestinal tract, breast, lung, and ovaries [44]. F-NETs associated with carcinoid syndrome and carcinoid heart disease had a heightened risk of thrombosis. The underlying mechanism may include the role of the increased serotonin levels in platelet activation [45]. In a study by Lopez-Vilchez et al., serotonin was shown to enhance the procoagulant activity of PLT, which engulfs TF-microparticles (TF-MVs) [46]. In this way, serotonin may enhance the PLT response by increasing thrombin generation, which plays a role in thrombus formation and can be significantly associated with increased cardiovascular risk [46]. According to Llobet D et al., human vesicle-associated membrane protein 8 (VAMP8) and serotonin transporter (SERT) may also be linked with platelet hyper-reactivity and VTE in a female Spanish population [47]. We found in our NET cohort that higher serotonin levels positively correlated with platelet levels, and an increased platelet count was associated with poorer survival.

AT-III acts as an endogenous serine protease inhibitor, and it plays a role in the inhibition of several enzymes related to the coagulation cascade, including thrombin [48]. Antithrombin deficiency is regarded as a risk factor for thrombosis in the general population, but its association with the risk of cancer-related thrombosis remains unclear. In an observational cohort study of 1127 cancer patients, AT-III was not associated with the risk of VTE; however, it showed a u-shaped association with the risk of all-cause mortality; patients with either very high or very low levels had worse OS [49]. We confirmed that AT-III levels were significantly lower in patients with metastatic disease compared to those without metastases, and lower levels of AT-III were observed in SI-NETs and were associated with increased levels of serotonin. We also noticed that patients during somatostatin analogues treatment had lower AT-III levels compared to those not receiving this treatment. However, further research is required in a larger population group to establish the impact of the applied treatment on coagulation parameters.

TF also leads to excessive thrombin generation and a subsequent hypercoagulable state [50]. It plays multiple roles, including its influence on cell migration and cell survival, interacting with integrins and receptor tyrosine kinases [51]. TF plays a critical role in the metastasis process [52,53]. Many types of cancer cells, including pancreatic, lung, gastric, breast, and brain cancer, were shown to express TF and release microparticles [TF-MVs] [44]. TF and TF-MVs correlated with poorer outcomes in cancer patients, especially those with pancreatic cancer [54]; however, no statistically significant association between TF and the risk of VTE was observed in all cases [55]. The downregulation of TF expression in Pan-NETs was shown to inhibit tumour cell proliferation in vitro. Additionally, the mTOR kinase inhibitor sapanisertib suppressed TF expression and activity, reducing Pan-NET growth in vivo [56]. TF was proposed as a potential molecule in cancer therapy for other types of cancer, e.g., tisotumab for the treatment of cervical cancer [53]. Given the promising reports on the role of TF in the survival assessment of patients with various cancers, including NETs, further studies are needed.

There are various available models for predicting the risk of VTE in cancer patients. Notably, these tools were neither developed nor validated for NETs, which exhibit many distinct features from common cancers. Since PP-S is intended to assess the risk of VTE for hospitalized or acutely ill patients, this may potentially result in their limited utility in NETs. Although Godinho et al. suggested that the ONCO-S may help stratify VTE risk in pancreatic cancers and identify patients who may benefit from thromboprophylaxis [57], in our study, this scale approached an AUC nearing the acceptable threshold, but it also did not meet the criterion and may require further research. The KS and new-VC-S can be useful for identifying patients at higher risk of mortality in other types of cancers [13]. Similarly, we found that higher scores on the KS or VC-S were linked to poorer survival in NET patients, which may relate to the location of the primary tumour, advanced clinical stage, and previously mentioned coagulation disorders that increase the risk of VTE in NET cases.

## 5. Limitations of the Study

This study has several potential limitations. The population size and the number of recorded VTE events (*n* = 10) were limited, which could impact the statistical multivariate power of the analyses. Additionally, follow-up was restricted to 12 months. Unknown cause of death, lack of long-term survival data (>12 months), and limited population size in relation to analyzed variables strongly limit the viability of survival analysis and require further studies and more extended observation periods. Furthermore, there is a possible underestimation of asymptomatic PE, as patients were not screened for asymptomatic PE cases due to the invasive nature of the computed tomography pulmonary angiography. Another aspect is the impact of the applied treatment, which also requires a more in-depth analysis of a larger population group. Finally, acute-phase proteins and family history, which may affect coagulation parameters, were not assessed. All the above warrant further research in this area.

## 6. Conclusions

Patients with NETs, particularly Pan-NETs, exhibited a higher risk of VTE. Plasma DD and fibrinogen levels may assist with predicting disease progression in NET patients. The use of the Khorana and Vienna CATS scales may be useful in NET patients Routine assessment of DD and fibrinogen may improve risk stratification for VTE in NET patients; however, extensive multicenter validation is necessary for clinical implementation.

## Figures and Tables

**Figure 1 cancers-17-03405-f001:**
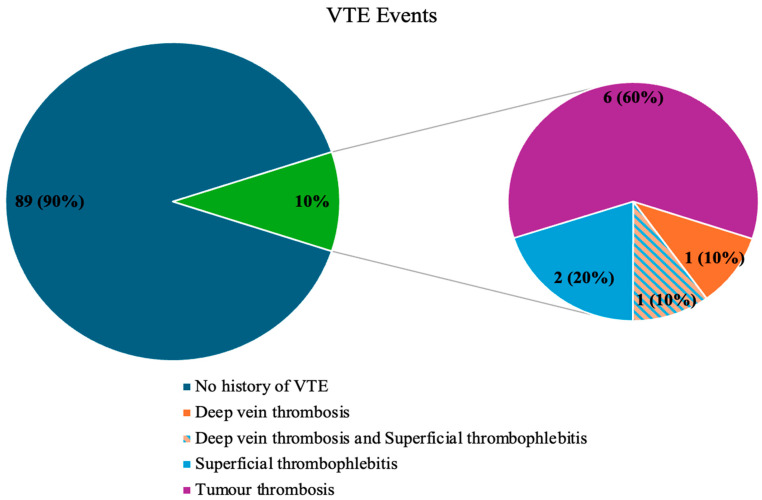
The occurrence of VTE events in the study group.

**Figure 2 cancers-17-03405-f002:**
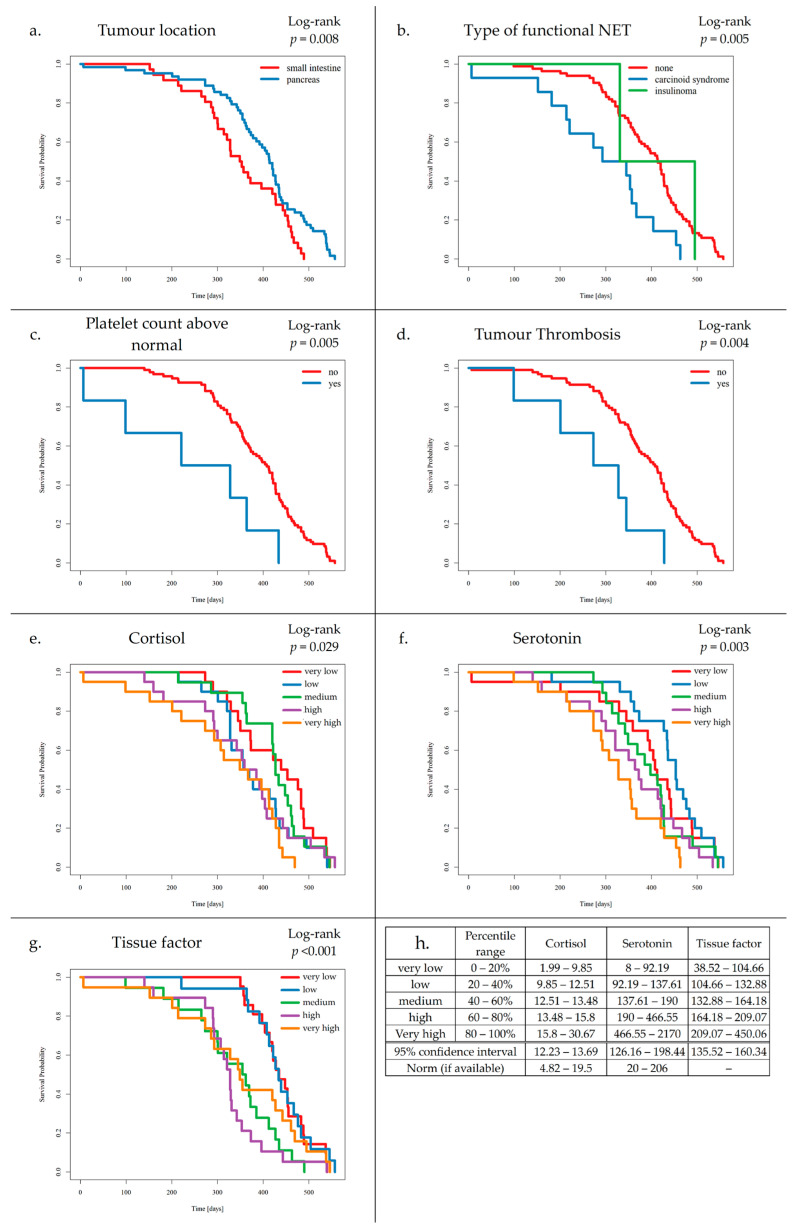
Kaplan–Meier survival curves for tumour location (**a**), type of functional NET (**b**), platelet count above normal (**c**), tumour thrombosis (**d**), cortisol (**e**), serotonin (**f**), tissue factor (**g**), and range legend for categorized levels of cortisol, serotonin, and tissue factor, with 95% confidence interval range and polish norm range (**h**).

**Figure 3 cancers-17-03405-f003:**
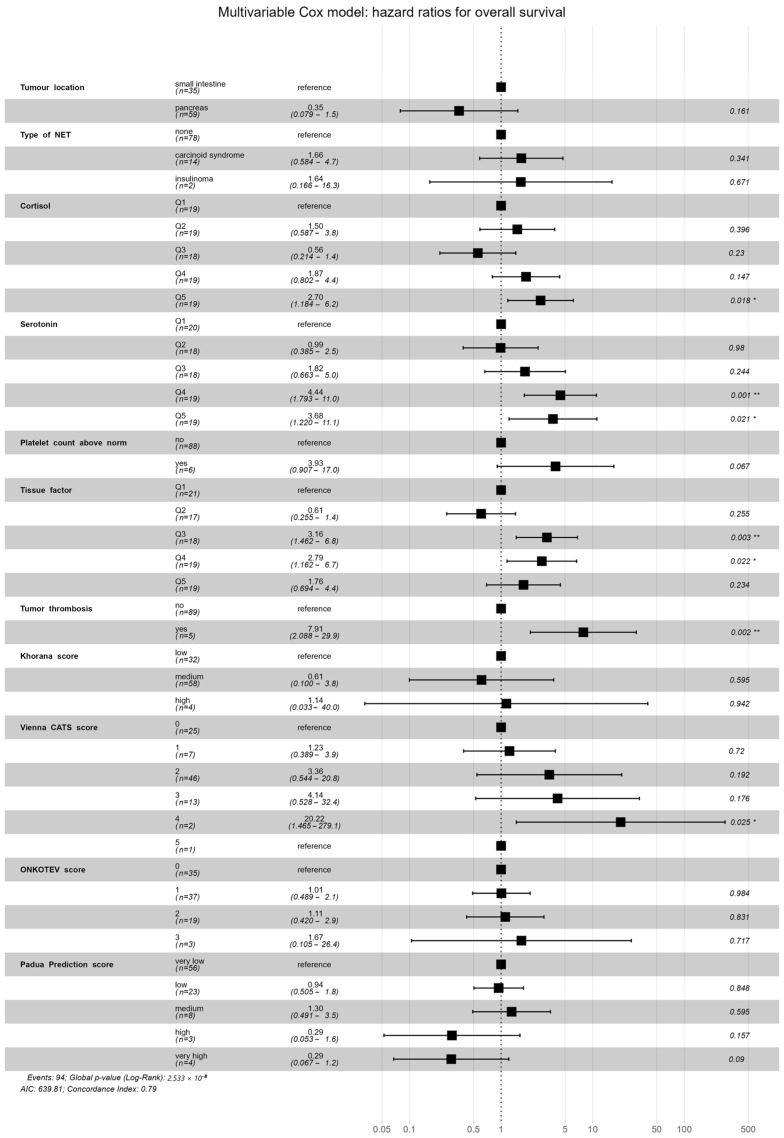
Multivariate analysis of selected parameters and VTE risk assessment scales in the Cox regression model. High levels of serotonin (Q4 *p* = 0.001, Q5 *p* = 0.021) and cortisol (Q5 *p* = 0.018), medium and high levels of TF (Q3 *p* = 0.003, Q4 *p* = 0.022), presence of TT (*p* = 0.002), and Vienna CATS score (score = 4, *p* = 0.025) were statistically significant in the specific Cox model results. Model concordance index = 0.79, model *p* < 0.001. Statistical significance is indicated as follows: ‘*’ for *p*-values lower than 0.05; ‘**’ for *p*-values lower than 0.005. The black square represents the hazard ratio, the black lines represent the hazard ratio 95% confidence interval, and the dotted line represents the hazard ratio of the reference (HR = 1.0).

**Figure 4 cancers-17-03405-f004:**
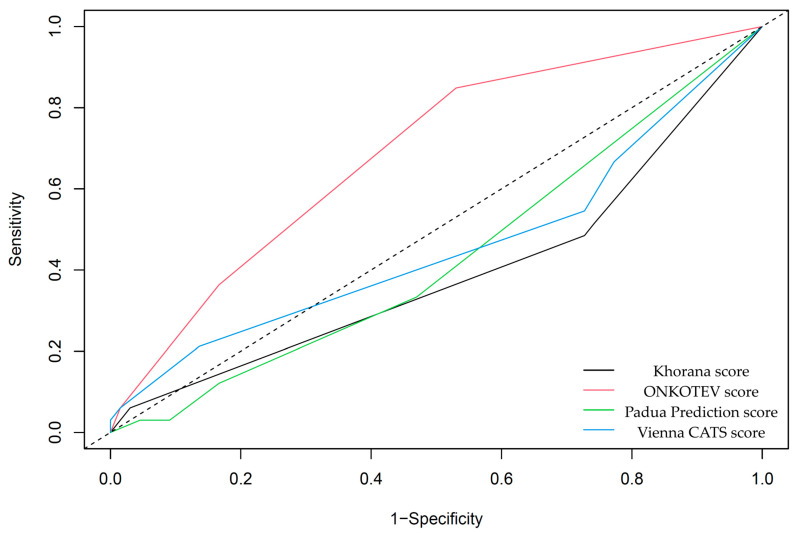
ROC analysis of VTE risk assessment scales and survival. None of the VTE risk assessment scales are considered good measures (AUC > 0.7) in the context of survival analysis. Among others, only the ONCOTEV score showed an AUC of 0.68 ± 0.05, *p* ≤ 0.001.

**Figure 5 cancers-17-03405-f005:**
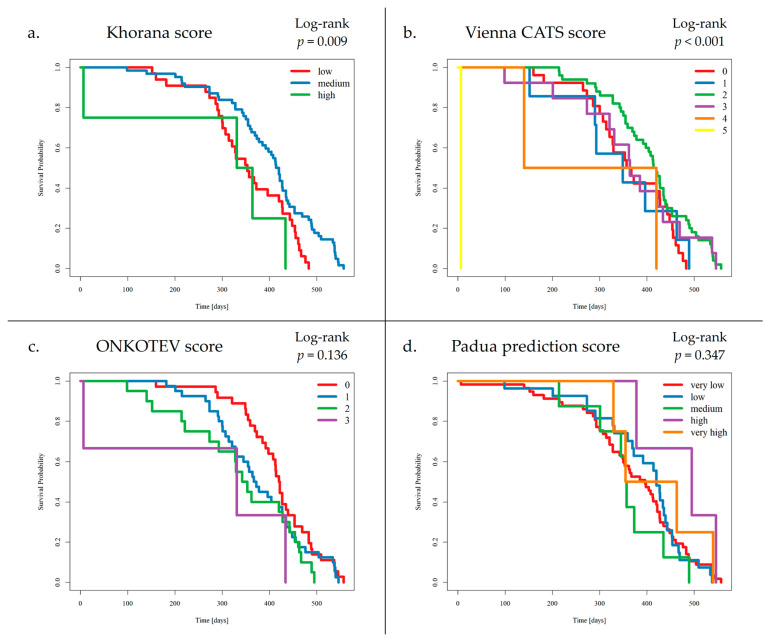
Survival analysis by VTE Risk Assessment Scales: (**a**) Khorana score, (**b**) Vienna CATS score, (**c**) ONCOTEV score, (**d**) Padua Prediction score. Patients who developed VTE events had a medium score on the KS (1–2 points) in 9 cases and a low score on the KS (0 points) in 1 case. Additionally, VTE events were detected in patients with 3 points, 2 points, and 1 point on the VC-S, in 3 cases, 5 cases, and 2 cases, respectively. A higher score on the KS or VC-S was associated with reduced survival (*p* = 0.009 and *p* < 0.001, respectively).

**Table 1 cancers-17-03405-t001:** The main inclusion and exclusion criteria for this study group.

The Inclusion Criterion	The Exclusion Criteria
Histologically confirmed diagnosis of a well-differentiated Pan-NET or SI-NET (NET G1 or G2)	Individuals under 18 years of age
	A history of any other cancers or blood disorders
	Heart or respiratory failure
	Acute cardiac or neurological events, recent surgeries or fractures (within the last month)
	Pregnancy
	Use of oral contraceptives
	Use of anticoagulants

Pan-NET = pancreatic neuroendocrine tumour, SI-NET = small intestinal neuroendocrine tumour.

**Table 2 cancers-17-03405-t002:** Characteristics of VTE Risk Assessment Scales. The table presents the parameters considered in assessing the risk scale.

Khorana Score ^1^	Vienna CATS Score ^2^	ONCOTEV Score ^3^	Padua Prediction Score ^4^
Cancer type: stomach, pancreas (2 score) lung, gynecologic, genitourinary excluding prostate (1 score) Body-mass index ≥ 35 kg/m^2^ (1 score) Platelet count ≥ 350 × 10^9^/L (1 score) Hemoglobin < 10 g/dL or using red blood cells growth factors (1 score) Leukocyte count > 11 × 10^9^/L) (1 score)	Khorana score (maximum 6 score) Soluble P-selectin ≥ 53.1 mg/mL (1 score) D-dimer ≥ 1.44 ug/mL (1 score)	Khorana score > 2 (1 score) Presence of metastatic disease (1 score) Compression of vascular/lymphatic structures (1 score) History of previous VTE (1 score)	Active cancer (patients with metastases to regional lymph nodes or with distant metastases who received chemotherapy or radiotherapy within the last 6 months) (3 score) Previous VTE (excluding superficial vein thrombosis) (3 score) Reduced mobility (3 score) Already known thrombophilic condition (3 score) Recent (≤1 month) trauma and/or surgery (2 score) Elderly age (≥70 years) (1 score) Heart and/or respiratory failure (1 score) Acute myocardial infarct and/or ischemic stroke (1 score) Acute infection and/or rheumatologic disorder (1 score) Body-mass index (BMI ≥ 30 kg/m^2^) (1 score) Ongoing hormonal treatment (1 score)

The interpretation of the scales: ^1^ Khorana score: 0 = a low risk, 1–2 = a medium risk, >2 = a high risk of VTE, ^2^ Vienna CATS score: ≥5 points = a high risk of VTE, ^3^ ONKOTEV score: 0–1 = a low risk, 2 = a medium risk, ≥3 = a high risk of VTE risk, ^4^ Padua Prediction Score: ≥4 = a high risk of VTE.

**Table 3 cancers-17-03405-t003:** Characteristic of the study and control group.

Parameters			Study Group *n* = 99		Control Group *n*= 47	*p* Value
Age (years)			57.63 ± 12.96		54.6 ± 12.57	0.119
Sex (female/male)			55/44		29/18	0.591
Body Mass Index (BMI)			25.98 ± 4.57		26.76 ± 4.29	0.389
Primary tumour location			Pancreas *n* (% of patients)	Small intestine *n* (% of patients)	-	-
			63 (63.6)	36 (36.4)	-	-
Histological grade	G1 (Ki-67 < 3%)		40 (40.4)	27 (27.2)	-	-
	G2 (Ki-67 3–20%)		23 (23.2)	9 (9.1)	-	-
Clinical staging	I		24 (24.2)	5 (5.1)	-	-
	II		11 (11.1)	0 (0.0)	-	-
	III		10 (10.1)	6 (6.1)	-	-
	IV		18 (18.1)	25 (25.3)	-	-
	Lymph node metastases		24 (24.2)	27 (27.2)	-	-
	Liver metastases		16 (16.1)	21 (21.2)	-	-
	Bone metastases		2 (2.0)	3 (3.0)	-	-
	Other metastases		4 (4.0)	12 (12.1)	-	-
Secretory status	Non-functioning		57 (57.6)	26 (26.3)	-	-
	Functioning		4 (carcinoid syndrome), 2 (insulinoma) (6.1)	10 (carcinoid syndrome) (10.1)	-	-
Treatment	Treatment-naïve		36 (36.4)	17 (17.2)	-	-
	Somatostatin analogues		17 (17.2)	17 (17.2)	-	-
	Molecular targeted treatment		1 (everolimus) and 1 (sunitinib) (2.0)	0 (0.0)	-	-
	Chemotherapy with capecitabine and temozolomide		0 (0.0)	1 (1.0)	-	-
Disease progression	Progressive		15 (15.2)	10 (10.1)	-	-
	Stable		48 (48.5)	26 (26.3)	-	-
VTE Risk Assessment Scales	Khorana score	Low (0 score)	0 (0.0)	32 (32.3)	0 (0.0)	-
		Medium (1–2 score)	58 (58.6)	4 (4.0)	0 (0.0)	
		High (>2 score)	4 (4.0)	0 (0.0)	0 (0.0)	
	Vienna CATS score	0 score	0 (0.0)	25 (25.3)	31(66.0)	-
		1 score	0 (0.0)	7 (7.1)	16 (34.0)	
		2 score	47 (47.4)	4 (4.0)	0 (0.0)	
		3 score	13 (13.1)	0 (0.0)	0 (0.0)	
		4 score	2 (2.0)	0 (0.0)	0 (0.0)	
		5 score	1 (1.0)	0 (0.0)	0 (0.0)	
	ONCOTEV score	Low (0–1 score)	49 (49.5)	27 (27.3)	0 (0.0)	-
		Medium (2 score)	11 (11.1)	9 (9.1)	0 (0.0)	
		High (≥3 score)	3 (3.0)	0 (0.0)	0 (0.0)	
	Padua Prediction Score	0 score	35 (35.4)	22 (22.2)	32 (68.1)	-
		1 score	19 (19.2)	8 (8.1)	12 (25.5)	
		2 score	4 (4.0)	4 (4.0)	3 (6.4)	
		3 score	3 (3.0)	0 (0.0)	0 (0.0)	
		≥4 score	2 (2.0)	2 (2.0)	0 (0.0)	
VTE events	Deep vein thrombosis		1 (1.0)	0 (0.0)	0 (0.0)	-
	Deep vein thrombosis and Superficial vein thrombosis		0 (0.0)	1 (1.0)	0 (0.0)	
	Superficial vein thrombosis		1 (1.0)	1 (1.0)	0 (0.0)	-
	Tumour thrombosis		6 (6.1)	0 (0.0)	0 (0.0)	-

**Table 4 cancers-17-03405-t004:** Characteristic of the patients with VTE events.

VTE Events	Primary Tumour Location	Histological Grade	Clinical Staging	Secretory Status	Disease Progression	Treatment	Time from Diagnosis <6 Months	DD Levels [µg/L, n: <500 µg/L]
Deep vein thrombosis	Pan-NET	G2	III	NF	no	Treatment-naïve	yes	680
Deep vein thrombosis and Superficial vein thrombosis	SI-NET	G1	IV	NF	yes	Treatment-naïve	yes	3676
Superficial vein thrombosis	Pan-NET	G2	IV	NF	yes	Treatment-naïve	yes	3382
	SI-NET	G2	III	NF	no	Treatment-naïve	no	313
Tumour thrombosis (splanchnic vein thrombosis)	Pan-NET	G1	IV	NF	yes	Somatostatin analogue, Everolimus	yes	590
	Pan-NET	G2	III	F (carcinoid syndrome)	yes	Treatment-naïve	yes	477
	Pan-NET	G2	III	NF	yes	Treatment-naïve	yes	198
	Pan-NET	G1	III	NF	yes	Treatment-naïve	yes	1651
	Pan-NET	G1	IV	NF	yes	Treatment-naïve	yes	393
Tumour thrombosis (portal vein thrombosis)	Pan-NET	G1	IV	NF	yes	Treatment-naïve	yes	2331

VTE = venous thromboembolism, DD = D-dimer, Pan-NET = pancreatic neuroendocrine tumour, SI-NET= small intestinal neuroendocrine tumour, NF = non-functioning, F = functioning.

**Table 5 cancers-17-03405-t005:** The results of the coagulation parameters in the study and control groups.

Parameters [Unit]	Study Group	Control Group	*p* Value
	Mean ± SD	Mean ± SD	
DD [µg/L]	957.59 ± 2021.86	400.26 ± 230.55	0.007
Fibrinogen [mg/dL]	318.98 ± 78.74	303.40 ± 55.45	0.301
PLT [10^9^/L]	255.28 ± 97.22	256.11 ± 54.93	0.435
AT-III [%]	101.71 ± 14.95	102.47 ± 11.34	0.795
TF [pg/mL]	157.71 ± 63.96	161.77 ± 25.56	0.160

DD = D-dimer, PLT = platelets, AT-III = antithrombin-III activity, TF = tissue factor.

**Table 6 cancers-17-03405-t006:** The results of the coagulation parameters in the Pan-NETs and SI-NETs groups.

Parameters [Unit]	Pan-NETs	SI-NETs	*p* Value
	Mean ± SD	Mean ± SD	
DD [µg/L]	1023.92 ± 2371.65	841.50 ± 1208.51	0.558
Fibrinogen [mg/dL]	311.73 ± 65.72	331.67 ± 97.21	0.302
PLT [10^9^/L]	260.56 ± 97.92	246.03 ± 96.65	0.430
AT-III [%]	104.63 ± 14.56	96.58 ± 14.42	0.010
TF [pg/mL]	156.36 ± 69.53	159.99 ± 54.16	0.474

Pan-NETs = pancreatic neuroendocrine tumours, SI-NETs = small intestine neuroendocrine tumours, DD = D-dimer, PLT = platelets, AT-III = antithrombin-III activity, TF = tissue factor.

**Table 7 cancers-17-03405-t007:** The correlations (r_S_) between coagulation parameters and selected clinical parameters.

Parameters	Age		WHO/ECOG Scale		Clinical Staging		Tumour Size		Lymph Node Metastases		CgA		Serotonin		5-HIO	
	r_S_	*p*	r_S_	*p*	r_S_	*p*	r_S_	*p*	r_S_	*p*	r_S_	*p*	r_S_	*p*	r_S_	*p*
**DD**	**0.38**	**<0.001**	**0.51**	**<0.001**	**0.32**	**0.001**	**0.23**	**0.022**	**0.25**	**0.014**	**0.32**	**0.001**	0.05	0.657	0.06	0.578
**Fibrinogen**	**0.23**	**0.021**	**0.21**	**0.040**	0.15	0.128	0.16	0.127	−0.04	0.675	0.15	0.127	0.15	0.136	0.15	0.129
**PLT**	0.02	0.831	0.07	0.487	0.08	0.432	−0.03	0.744	−0.01	0.951	−0.16	0.107	**0.22**	**0.029**	−0.13	0.203
**AT-III**	−0.01	0.897	0.00	0.998	**−0.22**	**0.031**	−0.15	0.130	**−0.20**	**0.044**	0.05	0.615	**−0.21**	**0.041**	−0.04	0.659
**TF**	0.12	0.238	**0.23**	**0.024**	**0.21**	**0.043**	0.10	0.323	0.19	0.074	0.05	0.632	0.00	0.983	0.14	0.165

DD = D-dimer, PLT = platelets, AT-III = antithrombin-III activity, TF = tissue factor, CgA = chromogranin A, 5-HIO = urinary 5-hydroxyindoleacetic acid, in bold: Spearmans correlation coefficients with *p* value lower than 0.05. DD showed a positive correlation with age in both the study and control groups (rS = 0.38, *p* ≤ 0.001 vs. rS = 0.35, *p* = 0.016, respectively). DD correlated positively with the WHO/ECOG scale (rS = 0.51, *p* ≤ 0.001), clinical staging (rS = 0.32, *p* = 0.001), and tumour size (rS = 0.23, *p* =0.022), lymph node metastases (rS = 0.25, *p* = 0.014). Among the biochemical markers, there was a positive correlation between DD and CgA (rS = 0.32, *p* = 0.001). Fibrinogen showed a positive correlation with age in the study group (rS **=** 0.23, *p* = 0.021). Fibrinogen concentration showed a positive correlation with the WHO/ECOG scale (rS = 0.21, *p* = 0.040). A positive trend between fibrinogen and serotonin levels was found, although it was not significant (rS = 0.15, *p* = 0.136). A positive correlation was observed between PLT and serotonin levels (rS = 0.22, *p* = 0.029). A negative correlation was found between AT-III and clinical staging (rS = −0.22, *p* = 0.031), as well as metastases to lymph nodes (rS = −0.20, *p* = 0.044) and serotonin levels (rS = −0.21, *p* = 0.041). Additionally, TF positively correlated with both WHO/ECOG scale (rS = 0.23, *p* = 0.024) and clinical stage (rS = 0.21, *p* = 0.043).

**Table 8 cancers-17-03405-t008:** ROC analysis of VTE risk assessment scales and survival.

VTE Risk Assessment Scales	AUC ± SD	*p*-Value
Khorana score	0.39 ± 0.05	0.059
Vienna CATS score	0.46 ± 0.06	0.501
ONCOTEV score	0.68 ± 0.05	<0.001
Padua Prediction Score	0.43 ± 0.05	0.191

AUC = area under the curve; None of the VTE risk assessment scales are considered good measures (AUC > 0.7) in the context of survival analysis. Among others, only the ONCOTEV score showed an AUC of 0.68 ± 0.05, *p* ≤ 0.001.

## Data Availability

The data used to support the findings of this research are available upon request from the corresponding author.

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
