# Peer review of "The Clinical Utility of Selected Coagulation Parameters in Predicting the Risk of Venous Thromboembolism in Neuroendocrine Tumours: A Prospective, Single-Centre Study"

_cancers, 2025, doi:10.3390/cancers17213405_

Round 1

Reviewer 1 Report

Comments and Suggestions for Authors

This paper examines a therapeutically significant but underexplored issue. This study examines the risk of venous thromboembolism (VTE) in individuals with neuroendocrine tumours (NET). There are no distinct thrombotic risk models or preventive advice in this area. This study is not only new and well-organised, but it also offers clear tables and statistics that are easy to read from a methodological point of view. This problem fits closely with MDPI's goals for cancer.

Nonetheless, the inquiry has several deficiencies that might be rectified by offering clarification, elaborating on the data, or restructuring the material prior to its acceptance.

The research examines a unique therapeutic issue: the classification of deep vein thrombosis risk in NETs, using four existing thromboembolic risk models and the gathering of prospective data.
The prospective design, the inclusion of healthy controls, the standardisation of coagulation tests, and the adherence to ethical norms all enhance trustworthiness.
Tables 3–7 with graphics like Kaplan–Meier curves, ROC, and Cox regression provide a short summary of the data. It has been shown that a correlation exists between coagulation markers and NET features.
This discussions does a great job of showing how limited the existing risk prediction methods for NETs are and putting the results in the context of the relevant studies.

Points nedded improvement

Abstract:

The abstract has a lot of numbers in it, even if it is neatly written.
It is best to shorten the data to highlight the most important findings, such the links between D-dimer, fibrinogen, Vienna CATS, and the Khorana scale.

Please explain why the sample size (n=99 NETs, n=47 controls) is what it is, and please note that no VTE risk scale was able to reach an area under the curve (AUC) greater than 0.7 when it came to predicting accuracy.

Introduction

It is possible that the explanation might be enhanced by offering a more comprehensive and lucid elucidation of the reasons why conventional risk models (Khorana, Vienna CATS) may not apply to NETs.

You might state something like, "Primary objective: assess the predictive validity of risk assessment scales; secondary objective: examine coagulation biomarkers as indicators of VTE." This is one way you may make the main and secondary objectives more clear.

Because References [5,6] are similar to studies done by the same organisation before, I would want you to provide me a full explanation of the new facts in the 2023 Cancers research.

Materials and methods

Please explain how you came up with the sample size or the rationale behind it, given you haven't done so before.

Do the results of the Doppler test back up what the radiologist found? Please explain more about the diagnostic criteria for venous thromboembolism (VTE) so that we can comprehend them better.

Tell us how you got the participants (volunteers, hospital personnel, etc.) and whether or not the control individuals were the same age, gender, and body mass index (BMI).

This can lower the power, hence there has to be statistical proof that continuous variables can be divided into quintiles, which are defined as "five groups, each with 20%."

For the sake of statistical analysis, could you kindly provide a full account of the methods that were utilised to update multivariate models? These methods took into account things like age, cancer stage, and metastasis.

There are no standards given, however the implementation of missing data management is discussed.

Results

Even if the results are easy to understand, the large number of p-values may make it hard to see the most important discoveries.
You should focus on the most significant facts, such fibrinogen, D-dimer, tissue factor, and platelet count.

Using bigger fonts and simpler legends might make the Kaplan–Meier and ROC figures clearer.

Since Table 4 has both instances, I would want to know whether the VTE occurrences happened before or during the follow-up period.

If it's acceptable, provide a composite endpoint analysis that includes things like extremely deep vein thrombosis, tumour thrombosis, and death.

Discussion

Even if the discourse provides a lot of background material, it may be better with a more thorough critical examination.

Examine the factors that influence the relationship between fibrinogen and D-dimer levels and the progression of the disease, which are not directly linked to the occurrence of incident venous thromboembolism (VTE).

Talk about why there were no pulmonary embolisms in the group. Could it be because there wasn't enough screening?

There should be more knowledge regarding the possible mechanistic links, which might involve serotonin, hormones produced by the NET, and procoagulant microparticles.

It is crucial to acknowledge the possibility for drugs such as mTOR inhibitors and somatostatin analogues to induce misunderstanding about the patient's status.

It is crucial to assess the external validity of the findings, namely their applicability to people beyond the Polish demography.

Limitation

Even if it is allowed, it can be helpful to include:

"Absence of long-term survival data (>12 months)" with the "monocentric design."

"Possible underestimation of asymptomatic pulmonary embolism."

"The multivariate power is limited by the small number of VTE events (n=10)."

Conclusions

It is conceivable that the text, although being well-composed, may have a more significant impact. "Routine assessment of D-dimer and fibrinogen may improve risk stratification for VTE in NET patients; however, extensive multicenter validation is necessary for clinical implementation."

Figures and tables

Make sure that the table numbering and figure positioning follow the rules set forth in the MDPI manual.

Make sure that each table only has one occurrence of each abbreviation and that any minor repetitions are mentioned.

Figure 1's caption may be made easier to understand since it repeats the main point that was made before.

Reviewer 2 Report

Comments and Suggestions for Authors

The overall study employs robust methods and is well-written. I have some comments that might improve the manuscript.

1. The study only follow-up time is only 12 months, which might not analyze late thrombotic events in slowly progressing NETs.

2.  The scores should also be reported for mortality. The interpretation conflates risk of VTE with prognosis.

3. The causal mechanisms of progression and mortality are not well explored.

4. Table 3 and Table 5 have inconsistent alignment and decimal separators.

5. Figure 2 should include a 95% CI for clarity.

Reviewer 3 Report

Comments and Suggestions for Authors

The authors presented a retrospective, observational study on the clinical utility of blood measurement of selected parameters of coagulation and compare the efficacy of various VTE risk assessment scales in patients with NETs. The study group is rather limited and, in turn, VTE incidents are low (n=10, < 10%).

Several corrections are suggested and listed below:

Lines 53-61: this is a common knowledge. Please provide the introduction for the interesting topic you present based on the potential mechanisms involved in the elevation of the DVT risk in NETs.

Lines 63: Please state in which way they are different?

Table 1 Would you consider family history as a risk factor?

Why did you choose those particular scores (line 85)? Are they in any way disease-oriented (oncological ones?)? Did you consider other scores e.g. Caprini in case of surgical procedures? Please comment on that in methodology.

Line 125: please define control group (who were healthy volunteers?).

Line 126: verb is missing (e,g. were noted?).

Line 127: please provide the range of follow up.

Please correct the numer of pages starting from the subparagraph 3.2 Selected Blood parametres of coagulation, together with respective lines.

Page 1/22 3.3 surivival analysis.., line 60: please furhter comment why the cause of death was unknown. How were these data collected? National registry?

Please add the respective section in the beginning of discussion stating why your paper is so unique, describing the major findings.

Round 2

Reviewer 3 Report

Comments and Suggestions for Authors

The authors responded to all the queries sufficiently.